# The Development of an Enhanced Recovery Protocol for Kasai Portoenterostomy

**DOI:** 10.3390/children9111675

**Published:** 2022-10-31

**Authors:** Peggy Vogt, Renee Tolly, Matt Clifton, Tom Austin, Joelle Karlik

**Affiliations:** 1Department of Pediatric Anesthesia and Pain Medicine, Egleston Hospital, Children’s Healthcare of Atlanta, 1405 Clifton Rd, Atlanta, GA 30322, USA; 2Department of Anesthesia, Emory University, Atlanta, GA 30322, USA; 3Department of Surgery, Emory University and Children’s Healthcare of Atlanta, 1405 Clifton Rd, Atlanta, GA 30322, USA; 4Department of Anesthesiology, University of Florida, Gainesville, FL 32611, USA

**Keywords:** pediatric surgery, Kasai procedure, pediatric anesthesia, enhanced recovery, enhanced recovery after surgery (ERP), regional anesthesia

## Abstract

Balancing post-operative adequate pain control, respiratory depression, and return of bowel function can be particularly challenging in infants receiving the Kasai procedure (hepatoportoenterostomy). We performed a retrospective chart review of all patients who underwent the Kasai procedure from a single surgeon at Children’s Healthcare of Atlanta from 1 January 2018, to 1 September 2022. 12 patients received the Kasai procedure within the study period. Average weight was 4.47 kg and average age was 7.4 weeks. Most patients received multimodal pain management including dexmedetomidine and/or ketorolac along with intravenous opioids. A balance of colloid and crystalloids were used for all patients; 57% received blood products as well. All patients were extubated in the OR and transferred to the general surgical floor without complications. Return of bowel function occurred in all patients by POD2, and enteral feeds were started by POD3. One patient had a presumed opioid overdose while admitted requiring a rapid response and brief oxygen supplementation. Simultaneously optimizing pain control, respiratory safety, and bowel function is possible in infants receiving the Kasai procedure. Based on our experience and the current pediatric literature, we propose an enhanced recovery protocol to improve patient outcomes in this fragile population. Larger, prospective studies implementing an enhanced recovery protocol in the Kasai population are required for stronger evidence and recommendations.

## 1. Introduction

The Kasai procedure is a portoenterostomy that is performed in infants with a diagnosis of biliary atresia, a life-threatening hepatic disorder which prevents bile drainage from the liver. The Kasai procedure takes several hours to perform, and may require significant intraoperative fluids, vasopressors and blood transfusion requirements in an infant with baseline fragile physiology. Postoperatively, balancing adequate pain control, respiratory depression, and return of bowel function is particularly challenging. The opioids required for adequate pain control may cause respiratory depression, prolonged intubation, limit return of bowel function, and lengthen admission duration [1,2].

Enhanced Recovery Protocols (ERPs) have gained popularity over the last decades. ERPs provide a perioperative evidence-based multi-step pathway designed to achieve early recovery for patients [3]. ERPs are gaining popularity in the pediatric surgery field but are still limited. Multiple reviews of the pediatric literature showed that despite a paucity of literature, ERPs were associated with favorable post-operative outcomes [4,5,6].

There is very limited data on the use of an ERP protocol for Kasai portoenterostomy patients. Potentially applicable Enhanced Recovery After Surgery (ERAS) Society protocols include a neonatal bowel and adult liver surgery pathways [3]. Epidural anesthesia, a regional anesthesia technique to limit opioid use, has been shown to shorten extubation times and hospital stays in Kasai patients [7]. Early extubation has been shown to be safe in similar populations including infants receiving pediatric liver transplant [8,9,10]. In addition, early extubation in liver transplants was shown to decrease pediatric intensive care unit (PICU) stays and overall hospital stays [8]. Therefore, these techniques seem appropriate for extrapolation for a Kasai ERP.

In this case, series, we aim to summarize our experience with Kasai procedures from January 2018 to June 2022. Based on our experience, we propose an ERP to improve patient outcomes in this fragile population.

## 2. Materials and Methods

After approval by our Institutional Review Board, we performed a retrospective chart review of all patients who underwent the Kasai procedure at Children’s Healthcare of Atlanta from 1 January 2018 to 1 September 2022. Recorded data included demographic information, ASA classification, fluid management, perioperative pain medications, intraoperative vasopressor use, hospital length of stay, and postoperative return of bowel function and feeding. All procedures were performed by one pediatric surgeon limiting confounding variables. An ERP protocol was developed from our patient data and extensive literature review.

## 3. Case Series

Twelve patients met the criteria for the study (M 3, F 9). All patients were classified as ASA 3. Average weight was 4.47 kg (range 3.675–5.43 kg) and average age was 7.4 weeks (range 5–11 weeks). Ethnic backgrounds were reported as black or African American (41.7%), Caucasian (16.7%), Hispanic or Latino (12%), black or African American/Caucasian (8.3%), and one declined (8.3%).

All patients were admitted prior to surgery and received preoperative dextrose containing intravenous fluids. Uniformly our patient population did not receive preoperative multimodal pain medications such as acetaminophen and gabapentin. Of the twelve patients, most (n = 8) had preoperative liver enzyme functions tests (LFTs) and all were abnormal. Two of the patients with preoperative LFTs did not have coagulation studies, one had normal coagulation, and the rest (n = 5) had elevated prothrombin time.

Intraoperative management was comparable including timely administration of antibiotics (cefazolin or cefoxitin), intraoperative temperature monitoring, and use of active warming strategies. All patients received rocuronium as the paralytic of choice (average dose 2.75 mg/kg). Dexamethasone administration was given to one third of the patients (n = 4, average dose 0.39 mg/kg). A third (n = 4) of the patients required intermittent phenylephrine boluses during the operation.

All patients received a balanced administration of both 5% albumin colloid (average 18.81 mL/kg) and crystalloid (average 32 mL/kg). Two-thirds of patients (n = 8) had their dextrose-containing fluids continued in the operating room. Most patients (n = 7) received packed Red Blood Cells transfusion (average 9.48 mL/kg).

Multimodal pain management was variable in our patient population. Under half of the patients (n = 5) received dexmedetomidine before extubation (average 0.44 mcg/kg). Ketorolac was only given in 1 patient. No patients received ketamine. Although all patients received fentanyl (average 4.73 mcg/kg), 75% were also supplemented with a long-acting opioid (hydromorphone: n = 6, average dose 0.011 mg/kg or morphine: n = 3, average dose 0.047 mg/kg). Only one quarter of our patients (n = 3) received a regional block (bilateral quadratus lumborum regional block, caudal, and thoracic epidural placement).

All patients were extubated in the operating room and transferred to the post-anesthesia care unit (PACU) hemodynamically stable on blow-by oxygen supplementation. None of the patients required ICU admission or had any surgical/anesthesia related complications in PACU. Most patients (n = 7) received a rescue pain medication (opioid n = 5, ketorolac = 2, acetaminophen = 1) before transferring to the floor.

Average total hospital days for our patients was 8.25 days (range 5–13). During admission, the patients remained on the surgical floor with no ICU transfers. In the majority of patients, TPN was started while awaiting return of bowel function and flatus (n = 11, 16.67% on POD 0, 66.67% on POD1, 8.33% on POD2). Enteral feeds were started after the return of bowel function in all patients (n = 1 on POD0, n = 1 on POD1, n = 7 on POD2, n = 3 on POD3). All patients had elevated post-operative liver function enzymes except for one.

One patient had a rapid response due to desaturations from presumed opioid overdose. The patient was placed on ½ L nasal cannula for 17 h before removal and did not require naloxone.

## 4. Discussion

Simultaneously optimizing pain control and bowel function while avoiding over sedation is possible in infants receiving the Kasai procedure. In our case series, we examined variables of known ERPs in our Kasai pediatric population and focused on post-operative outcomes. We considered length of stay as a primary goal for our enhanced recovery protocol recommendations. As secondary outcomes, we included parameters measuring return of bowel function, respiratory complications, and ICU transfers. Based on our experience and the current pediatric literature, we propose an ERP to improve patient outcomes in this fragile population (see Table 1).

Despite the advancement of adult ERPs, its implementation in the pediatric community is sparse. In 2016, a literature review reported only five studies examining the impact of ERP in all pediatric surgical patients [4]. By 2022, the number of studies had grown to sixteen, signaling interest in the field [6,30]. Currently, the ERAS Society has only one pediatric protocol focusing on neonatal bowel surgery but evidence is limited [17]. Gao et al. published an ERP for pediatric patients undergoing gastrointestinal surgery and found it reduced hospital cost, shortened length of stay and improved bowel recovery time in a patient population of mean 4.4 years [31]. Our study evaluated neonates, who are more prone to severe cardiovascular critical events perioperatively [32]. Since our analysis was retrospective, the average admission duration of 8.25 days and initiation of enteral feeds by POD 3 cannot definitely be attributed to an ERP. However, the lack of any critical perioperative event in our case series support the protocol.

The preoperative components of our proposed protocol focus on optimizing neonates for extended abdominal surgery. This infant population is particularly prone to hypothermia and hypoglycemia, both of which can cause significant hemodynamic and physiologic consequences [11,12]. To prevent hypothermia the OR should have a higher ambient temperature, warning lights, and forced air warmers. Glucose monitoring should occur perioperatively and dextrose containing fluids are recommended by our ERP. There is no consensus as to the amount of glucose containing solutions a neonate should receive therefore titration to each patient should be done by the discretion of the surgical and anesthesia teams [33]. The use of preoperative multimodal pain medication is limited in our population. Acetaminophen is not recommended due to the pre-existing transaminitis and coagulopathies in biliary atresia patients. The role of gabapentin for refractory pain in the neonatal intensive care unit is gaining popularity, however it is not FDA approved in children less than one year old and is not included in our ERP [13].

Fluid management was at the discretion of the anesthesiologist. We recommend avoiding copious fluid administration by placing IVFs on a pump with a programmed rate. Our patients received primarily crystalloids, following published adult hepatic surgery guidelines that strongly recommend the use of balanced crystalloid over 0.9% saline or colloids [34]. However, colloids are often used at our institution for hypovolemia when hyperchloremic metabolic acidosis is present and transfusion is not yet indicated; all patients received colloids as a result. A majority of our patients also required pRBCs for intraoperative blood loss. We strongly support the use of goal-directed transfusion guidelines to limit unnecessary exposure to blood products and avoid the high mortality associated with anemia [17].

Multimodal analgesia is a widely used technique to manage acute post-surgical pain and support early return of bowel function. In our case series, dexmedetomidine was the most commonly used non-opioid analgesic. Dexmedetomidine has been extensively studied in pediatric spinal procedures, intensive care units, procedural sedation for diagnostic procedures, and cardiac surgeries. It is an efficacious analgesic and reduces sympathetic tone, anxiety, and emergence delirium [24]. Given the neonatal population, we recommend utilizing dexmedetomidine after extubation to avoid unnecessary additional sedation. Intravenous ketorolac use has been associated with early discharge and decreasing inpatient opioid use in combination with other non-narcotic analgesics in pediatric surgeries [22,23]. Despite the limited use in our case series, we recommend IV ketorolac 0.5 mg/kg at surgical closure, in communication with the surgical team, and continued throughout the postoperative period every 6 h unless contraindicated [25].

Opioids are necessary for post-operative pain control but contribute to respiratory depression and decrease bowel motility. Intraoperatively, our patients were given short- and long-acting opioids as part of a balanced anesthetic and all patients were extubated in the OR. If possible, we strongly recommend completing these cases during daytime hours to ensure appropriate staffing and post-operative monitoring. Given the higher incidence of respiratory events in infants less than a year of age and ASA status ≥ 3, vigilance must be taken with the titration of opioids [35,36]. In this case, series, no PACU respiratory events were reported, however one patient had a presumed opioid overdose while on the floor requiring a rapid response and brief oxygen supplementation. To avoid such events, we recommend judicious use of opioids and ordering small doses of breakthrough medication as needed instead of scheduling opioids [17].

While only a few patients received a regional or neuraxial technique, we suggest abdominal wall regional or neuraxial blocks at the conclusion of the procedure. Regional/neuraxial anesthesia combined with non-opioid analgesics are proven to be advantageous in reducing opioid requirements and pain scores in other pediatric ERP studies and in the Kasai population [6,7,37]. However, concerns of coagulopathy associated with biliary atresia and prolonged hepatic surgery often prevent use of neuraxial techniques [34]. Other options can include truncal blocks or incision infusion catheters [34]. IV dexamethasone use has additional benefits to extend the duration of a sensory block, with moderate quality of evidence, which further strengthens our recommendation in our ERP [38]. If the patient received a regional/neuraxial block, we suggest a proper consultation with the acute pain service for appropriate follow up as well as continued monitoring of coagulation.

The components of our ERP protocol, such as normothermia and fluid management, are essential parts of perioperative neonatal care. However, unique to the Kasai procedure are the avoidance of acetaminophen given hepatic disease, consideration of coagulopathy with regional anesthesia, focus on multimodal pain control for a relatively larger surgical incision, and need for postoperative antibiotics given the ongoing infectious risk. Furthermore, the lack of knowledge of ERPs is a major barrier to implementation [39]. By incorporating standard neonatal anesthetic practices and highlighting Kasai procedure specific components, we hope to bring awareness of neonatal ERPs to expand utilization. 

While the limited number of Kasai procedures per year makes meaningful data collection challenging, a general trend of starting early enteral feeds (24–48 h) did exist in our patient population. In addition, we had no PACU respiratory events, no ICU admissions or transfers, and only one short-lived minor respiratory event on the floor. We believe these results support the use of an ERP protocol to promote multimodal analgesia, regional anesthesia, and OR extubation in this patient population.

Limitations for this case series include the retrospective design and limited number of eligible patients. As this case series is retrospective, we did not have an ERP in place at the time of operation. Collectively we are merely making suggestions from our data of twelve Kasai patients over the past three years. On review of our patients, we assessed similarities in management that may have contributed to the population’s success. These components were added to the ERP and then searched in the literature for confirmation. The components that were not widely used were researched within the literature, discussed with our perioperative team and will be implemented for future cases. Our data is provisional but will guide will future prospective studies for stronger conclusions and recommendations. The suggested ERP protocol should be evaluated to fit the needs of each institution. However, our ERP protocol is supported by the literature and our data in which a majority of patients received each component. We believe our ERP is a comprehensive plan to improve outcomes and morbidity in this fragile patient population.

## 5. Conclusions

We recommend implementing a basic ERP at our institute that consists of parental education, preoperative dextrose-containing fluids, monitoring temperature and maintain normothermia, antibiotics regimen, strict fluid management, early extubation, multimodal pain regimen with regional/neuraxial anesthesia, and early enteral feeds. Larger, prospective studies implementing many aspects of an enhanced recovery protocol with adequate controls in the Kasai pediatric surgical population are required for stronger evidence and recommendations.

## Figures and Tables

**Table 1 children-09-01675-t001:** ERP Protocol Component Compliance.

ERP Protocol Components	Number of Patients
*Pre-Operative Management* [11,12,13,14,15]	
Utilize dextrose-containing fluids the night previous to avoid hypoglycemia and hypovolemia	12/12
Pre-op dose of Tylenol not recommended given due to liver function and coagulation abnormalities	12/12
Pre-op gabapentin not recommended given the risk of sedation and lack of FDA approval in this age group	12/12
Pre-warm the room, ensure forced air warmer and warming lights are in the operating room	12/12
*Initial intra-operative period* [16,17]	
Administer stress dose steroids when indicated	12/12
Place nasopharyngeal or esophageal temperature probe and turn on forced air warmer, maintain normothermia (36.5 degrees)	12/12
Give dexamethasone 0.5 mg/kg to minimize airway edema and maximize likelihood of extubation	4/12
Muscle relaxants for surgical assistance	12/12
Administer antibiotics within 60 min prior to incision	12/12
*Fluid Management* [12,18,19,20]	
All maintenance fluids on pump during case	N/A
If present, replacement of pre-op fluid deficit at the discretion of the anesthesiologist caring for the patient.	12/12
If the need for transfusion of any blood products arises, it will be decided by the attending anesthesiologist and surgeon. Consider a Hb < 9 for a healthy full-term infant or Hb < 10–11 for infant with oxygen requirements	7/12
Recommend dextrose-containing fluids with intermittent glucose checks to avoid hypo- and hyperglycemia. Titrate fluids accordingly.	12/12
If hypotension with concern for hypovolemia and not meeting transfusion requirements, then consdier 10 mL/kg albumin bolus to avoid hyperchloremic acidosis and/or hyponatremia.	12/12
*Perioperative Pain Management* [21,22,23,24,25]	
Regional anesthesia: Consider epidural, caudal, or abdominal wall regional block (check coagulopathy if considering neuraxial)	3/12
IV Ketorolac at closure, communicate with surgical team for appropriate timing	1/12
Dexmedetomidine bolus at discretion of anesthesiology team. Consider dexmedetomidine AFTER extubation to avoid sedation	5/12
Attempt to use narcotics judiciously to maximize likelihood of extubation	12/12
If continued pain concerns, low dose ketamine bolus (0.5 mg/kg^−1^ mg/kg) but consider post-operative sedation	0/12
*Postoperative Care* [21,26,27,28,29]	
Pain team consult if patient received regional anesthesia	3/12
Scheduled ketorolac, q12hr	1/12
Consider lingual sucrose/dextrose for minor non-painful procedures such as NG tube placement	N/A
Continue perioperative antibiotics until tolerating enteral feeds. Switch to long-term trimethoprim/sulbactam when appropriate	12/12
Consider early enteral feeds within 24–48 h if appropriate; Breast milk preferred if available and parents approve	9/12

These are guidelines we are attempting to implement, of course patient safety is our primary concern, and clinical choices are up to each team caring for the individual patient.

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
