# Peer review of "The Development of an Enhanced Recovery Protocol for Kasai Portoenterostomy"

_children, 2022, doi:10.3390/children9111675_

Round 1
Reviewer 1 Report
- The authors proposed the Enhanced Recovery Protocol (ERP) for the Kasai procedure. They reviewed the experience of 12 patients who received the Kasai procedure. However, they didn't mention how they used the data to create/validate the ERP to shorten the length of stay. The authors should elaborate on how this is specific for Kasai as opposed to other pediatric abdominal surgeries. How was the data used to develop this ERP? Adding these to the methodology would help your readers understand how it was created based on evidence from the case series as well as the previous research.
Reviewer 2 Report
This is a retrospective study about the application of enhanced recovery protocol after Kasai portoenterostomy among pediatric patients with biliary atresia. However, the protocol suggested by the authors is a generalized treatment policy already implemented in most major centers. It is judged as not new content that suggests new academic value.
Round 2
Reviewer 1 Report
Since the authors could not show any benefit with the ERP with their current data, it is too soon to be published. It might lead the wrong way.
Reviewer 2 Report
Thank you for correcting the text in such a short period of time.
Reading the comments in response to the reviewer's question, this study deserves to be considered a useful study for institutions who do not familiar with neonatal ERPs associated with Kasai operation.
